# Review article: climate change impacts on dam safety

Javier Fluixá-Sanmartín[1], Luis Altarejos-García[2], Adrián Morales-Torres[3], and Ignacio Escuder-Bueno[4]

[1]Centre de Recherche sur l'Environnement Alpin (CREALP), Sion, 1951, Switzerland
[2]Department of Civil Engineering, Universidad Politécnica de Cartagena (UPCT), Cartagena, 30202, Spain
[3]iPresas Risk Analysis, Valencia, 46023, Spain
[4]Department of Hydraulic Engineering and Environment, Universitat Politècnica de València (UPV), Valencia, 46022, Spain

**Correspondence:** Javier Fluixá-Sanmartín (javier.fluixa@crealp.vs.ch)

**Abstract.** Dams as well as protective dikes and levees are critical infrastructures whose associated risk must be properly managed in a continuous and updated process. Usually, dam safety management has been carried out assuming stationary climatic and non-climatic conditions. However, the projected alterations due to climate change are likely to affect different factors driving dam risk. Although some reference institutions develop guidance for including climate change in their decision support strategies, related information is still vast and scattered and its application to specific analyses such as dam safety assessments remains a challenge.

This article presents a comprehensive and multidisciplinary review of the impacts of climate change susceptible to affect dam safety. The global effect can be assessed through the integration of the various projected effects acting on each aspect of the risk, from the input hydrology to the calculation of the consequences of the flood wave on population and assets at risk. This will provide useful information for dam owners and dam safety practitioners in their decision-making process.

*Copyright statement.* TEXT

## 1 Introduction

Large dams as well as protective dikes and levees are critical infrastructures whose failure has high economic and social consequences. Although usually very low, these infrastructures have an associated risk that must be properly managed in a continuous and updated process. In the dam safety context, risk can be estimated by the combined impact of all triplets of scenario, probability of occurrence and the associated consequence (ICOLD, 2003). Risk analysis is a useful methodology that encompasses traditional and state-of-the-art approaches to manage dam safety in an accountable and comprehensive way (Bowles, 2000; Serrano-Lombillo et al., 2013). The development and application of risk assessment techniques worldwide in the dam industry (ANCOLD, 2003; ICOLD, 2005; SPANCOLD, 2012; USACE, 2011b) has helped informing safety governance and supporting decision-making in the adoption of structural and non-structural risk reduction measures.

Most risk assessments in the past assumed a stationary condition in the variability of climate phenomena, including the frequency and magnitude of extreme events (National Research Council, 2009). However, changes in climate factors such as

variations of extreme temperatures or frequency of heavy precipitation events (CH2014-Impacts, 2014; IPCC, 2012b; Walsh et al., 2014) are likely to affect the different factors driving dam risks (Bowles et al., 2013a; USBR, 2014). The assumptions of stationary climatic baselines are no longer appropriate for long-term dam safety management (USACE, 2016). An update of risk components (loads, system response and consequences) to take into account the new climate change scenarios becomes imperative for adaptation and decision-making support under a more resilient approach.

In this context, some reference institutions (USACE, 2014; USBR, 2014, 2016) are actively developing and implementing guidance for including climate change in their decision support strategies (U.S. Government Accountability Office, 2013). In other cases, efforts have been done in the evaluation of climate change impacts on dam safety surveillance but further research is subjected to new findings and advances in the knowledge level (OFEV, 2014).

However, climate change related information is vast and scattered, and its application to specific analyses such as dam safety assessments remains a challenge for the dam engineering community. Although a considerable amount of research has been done so far, its application to current dam safety practice is still in prospect (Bahls and Holman, 2014) and needs to be done based on national and supranational overall adaptation plans (Commonwealth of Australia, 2015; European Commission, 2013; OECC, 2008). Moreover, the impacts of climate change effects on dam safety are usually analysed separately and aim at specific aspects. Most studies tend to focus only on the impact of climate change on the hydrological loads (Bahls and Holman, 2014; Chernet et al., 2014; Novembre et al., 2015) relegating or ignoring other aspects. Other studies with a wider scope only reach qualitative assessments (Atkins, 2013) limiting their applicability to screening analyses.

The global effect of climate change on dam risk must be assessed through the integration of the various projected effects acting on each aspect, taking into account their interdependencies, rather than by a simple accumulation of separate impacts. It is thus valuable to adopt a comprehensive approach to address climate change influence on dam safety management. In this context, dam risk models represent a useful basis on which such assessments can be structured.

In this work the authors seek a multidisciplinary and structured review of the most relevant impacts of climate change on the different dam safety components, from the input hydrology to the calculation of the downstream consequences of the inundation on population and assets at risk. In order to decompose such impacts on the different risk aspects, a risk analysis approach has been adopted. Moreover, practical techniques for their direct application are presented to provide useful information for dam owners and dam safety practitioners in their decision-making process.

## 2 Risk analysis approach for structuring climate change impacts

The effects of climate change are expected on a variety of factors affecting the dams, from the incoming floods to the definition of downstream consequences. Thus, in order to analyse the impacts of climate change on the global safety of a dam, it is necessary to decompose them in the different aspects that integrate the dam risk. Some techniques help addressing such analyses in a comprehensive way and structuring the way in which the risk assessment is envisaged.

In particular, risk analysis is a useful methodology to manage dam safety in an accountable and comprehensive way (Bowles et al., 2013b). Risk can be defined as the combination of three concepts: what can happen (infrastructure failure), how likely

is it to happen (failure probability), and what are its consequences (failure consequences) (Kaplan, 1997). Merz et al. (2010) propose a non-stationary definition of flood risk that includes damage and probability of occurrence. Based on these definitions, risk can be quantified with the equation set by Kaplan and Garrick (1981) and used extensively across different sectors in the industry (Altarejos-García et al., 2012; Aven, 2012; Serrano-Lombillo et al., 2011):

$$Risk = \int P(loads) \cdot P(response|loads) \cdot C(loads, response) \tag{1}$$

where the integral is defined over all the events under study, $P(loads)$ is the probability of the different load events, $P(response|loads)$ is the conditional probability of the structural response for each load event and $C(loads, response)$ are the consequences of the system response for each load event.

In this context, risk models are the basic tool used for the quantitative assessment of risk, integrating and connecting most variables concerning dam safety (Ardiles et al., 2011; Bowles et al., 2013a; Serrano-Lombillo et al., 2012). These models can be structured using influence diagrams such as the one presented in Figure 1 (SPANCOLD, 2012). Each node represents a variable related to each term of risk as defined in Eq. (1):

- **Loads of the system.** This term corresponds to the loads to which the dam will be subjected, and focuses on the upstream components of the dam. In particular, incoming floods are envisaged as the main hydrological load, and the rest of the component defines how the dam-reservoir system responds when confronted to such hydrological events.

- **System response (or failure probability).** This contains the information of the failure modes and the definition of the conditional probability of failure.

- **Consequences (economic, loss of life or any other).** This component includes an estimation of the consequences downstream of the dam for all the significant failure modes, including the dam break modelling.

In this work, the risk modelling approach shown in Fig. 1 has been chosen to structure and organize the assessment of all the potential impacts by disaggregating them on the different components of risk. The advantage of using this approach is threefold:

- The analysis is performed in a comprehensive way where the total risk and the climate change impacts are evaluated jointly, taking into account their interdependencies.

- All the risk components are evaluated, which avoids neglecting certain factors affecting the global safety.

- It is also possible to determine the contribution of each dam safety component to the overall risk impact, thus highlighting which is more susceptible to climate change or has more influence in the final risk level.

## 3 Climate change impacts on dam risk components

What follows is a review of the main climate change impacts on the dam risk components as presented in Sect. 2. The overall effect of climate change on risk can be assessed based on how these components are susceptible to change.

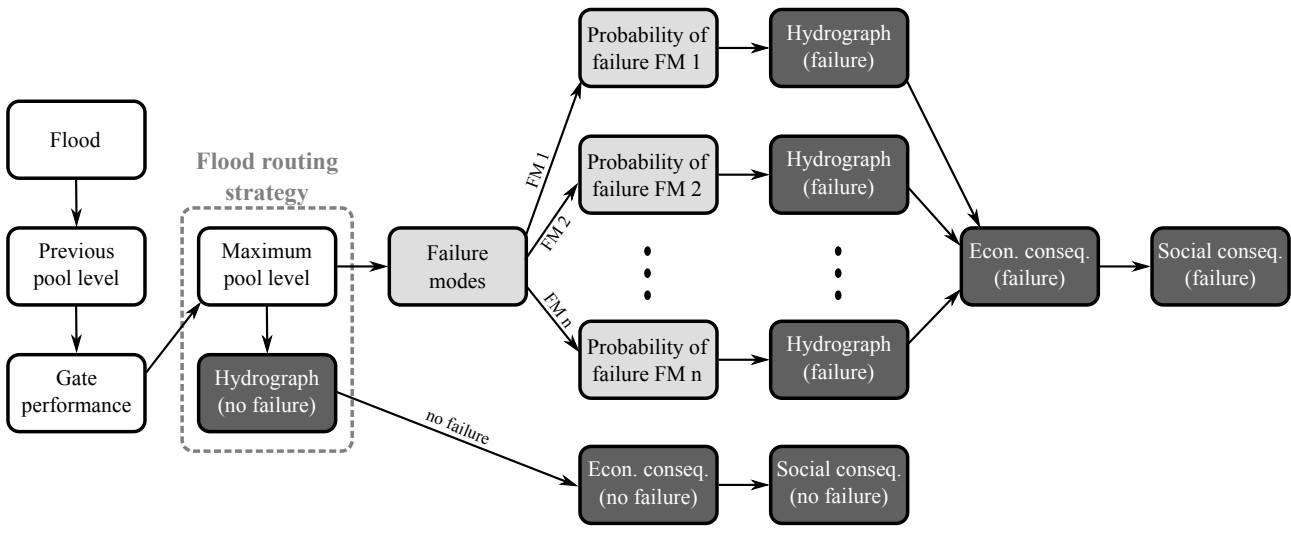

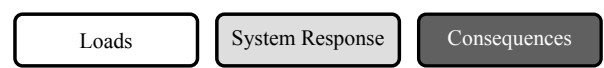

**Figure 1.** Standard risk model diagram for the hydrologic scenario, divided in Loads, System Response and Consequences nodes (adapted from SPANCOLD (2012)).

The present review focuses on the impacts of climate change on dam's safety under a hydrological scenario, which means that floods are the main load component to which the dam-reservoir system is subjected.

It is worth mentioning that risk impacts of climate change are conditioned by climatic but also by non-climatic drivers (IPCC, 2014) such as population increase, economic development, or water management adaptation. In certain cases, these non-climatic drivers may have a significant influence in the dam risk calculation and have been considered in the research.

Moreover, climate change is susceptible to impact both normal components (such as the population exposure downstream of the dam) and extreme components (such as the flood events) of risk, which can be captured by using the proposed risk analysis approach.

## 3.1 Loads of the system

### 3.1.1 Flood

In the hydrological scenario, floods are the initiating event (node) that creates the loads to which the dam is subjected and will be referred here as the upstream flow into the reservoir. The probabilities of the emerging branches are defined by the frequency occurrence linked to the inflow hydrographs (Fig. 2 (a)), introducing the temporal component to the risk calculation [consequences/year]. These are associated with a given return period (T) or its equivalent annual exceedance probability (AEP).

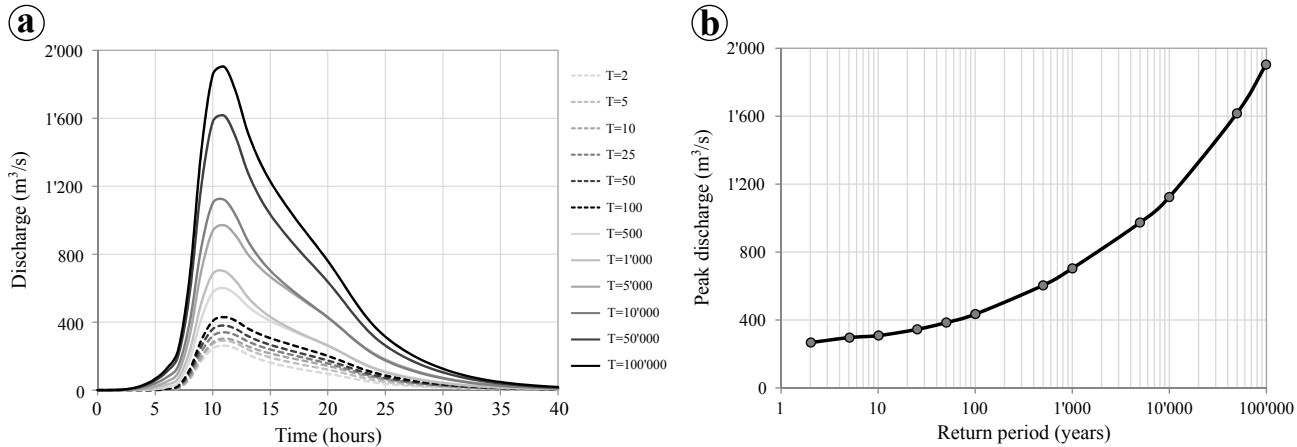

**Figure 2.** (a) Example of upstream hydrographs as used in the flood routing computation. (b) Resulting flood frequency characterization of the maximum values of peak discharge ($Q_P$), used in the Flood node.

Different analyses can be performed to estimate the occurrence probability of these events using deterministic, parametric, probabilistic and stochastic methods (World Meteorological Organization, 2008). Some of them seek relating the magnitude of one or more hydrological variables with T. A widely used approach to characterize this relation is to perform frequency analyses of the maximum values of peak discharge ($Q_P$) and/or volume (V) (Fig. 2 (b)): while univariate analyses focus on the individual influence of each factor, multivariate analyses are used to obtain their joint distribution in order to know the probability of occurrence of a given inflow hydrograph (Requena et al., 2013; Serinaldi and Grimaldi, 2007; Zhang and Singh, 2006). The main component of dam safety affected by climate change is the hydrology of river basins defined by the incoming floods. Heavy precipitation has an important influence, but floods are also affected by other factors including snow cover and snowmelt (Arheimer and Lindström, 2015; Fassnacht and Records, 2015), vegetation or soil moisture (Mostbauer et al., 2017). Changes already identified in these factors are likely to modify the characteristics of floods, namely their magnitude and/or frequency (IPCC, 2014).

The assessment of the correspondence between changes in climate factors and flood occurrence remains however complex. Although there are abundant studies on the changes and trends for rivers over the past years (Hannaford and Marsh, 2008; Petrow and Merz, 2009; Villarini et al., 2009), there is still a lack of evidence regarding patterns in the magnitude and/or frequency of floods on a global scale (IPCC, 2012b). Nevertheless, physical reasoning suggests that projected variations in heavy rainfall and other factors in some catchments or regions would contribute to variations in local floods (Bates et al., 2008; Kundzewicz et al., 2007). Existing analyses of flood changes at the basin scale (Prudhomme and Davies, 2009; Raff et al., 2009; Taye et al., 2011) justify the need for a re-evaluation of flood frequency and magnitude impacting dam safety. To take into consideration the non-stationarity hypothesis in flood frequency analysis, some works apply methods to account for the effects of climate change in flow regimes (Gilroy and McCuen, 2012; López and Francés, 2013).

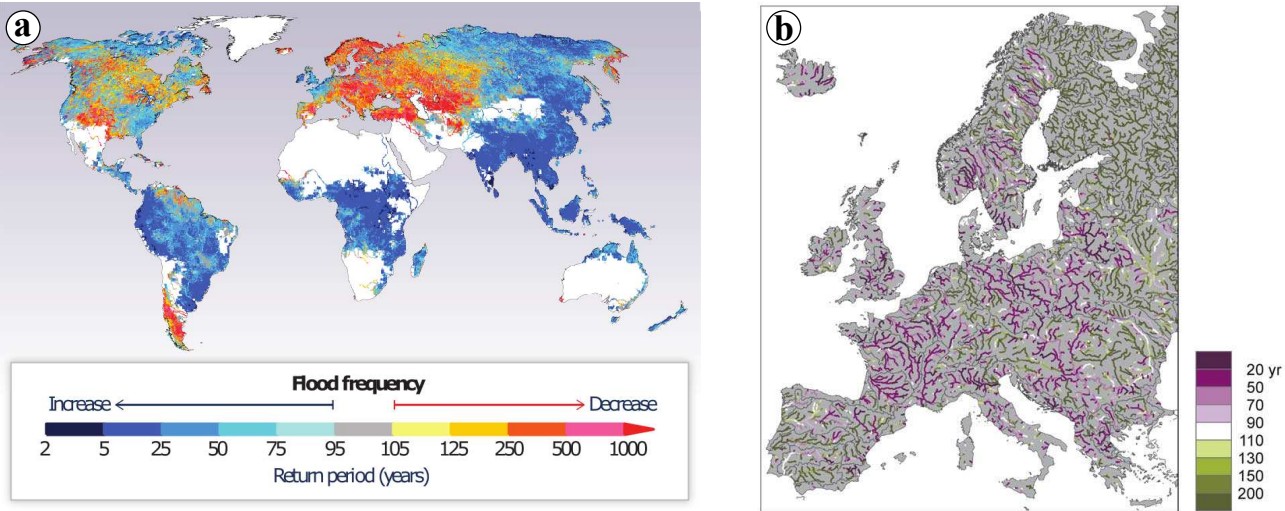

**Figure 3.** (a) Multi-model median return period (years) in the 2080s for the 20[th] century 100-year flood based on one hydrological model driven by 11 GCMs under RCP8.5 (from Hirabayashi et al. (2013)). (b) Change in recurrence of a 100-year flood in Europe in H12A2 scenario run using a Gumbel distribution (from Dankers and Feyen (2008)).

Direct analyses on the expected changes in flood's frequency and/or magnitude can be applied using existing studies of the matter on the study region. For instance, the works of Dankers and Feyen (2008), Hirabayashi et al. (2013) or Wobus et al. (2017) present the expected variations of characteristic floods in magnitude or frequency at large scales (Fig. 3). These variations can be then applied to the concerned floods of the basin by a simple extrapolation of the hydrographs based on the
5 ratio between their peaks.

More specific analyses require to rely on local effects on floods (at catchment-wide scale) rather than apply regional- or continental-scale findings. When no detailed information is available at the catchment level, site-specific analyses are required. Most studies use adapted global (GCMs) and regional (RCMs) climate models coupled with hydrological and land surface models to assess how floods are expected to change at the watershed level (Chernet et al., 2014; Duan et al., 2017; Khazaei
10 et al., 2012). Climate models can be applied to present or historical climatic variables (mainly precipitation and temperature) in order to obtain projections of future climate series (preferably at daily or sub-daily time steps). These new series are then introduced as inputs to the hydrological model. The resulting flows are then statistically analysed (the longer the simulation period, the more accuracy) to derive the flood frequency statistics.

In most cases climate change projections from GCMs cannot be directly used because their spatial resolution is too coarse for
15 modelling the hydrological processes at the required regional or even local scale, and thus must be downscaled and eventually bias-corrected. A synthetic diagram of a common methodology for the frequency analysis of floods as used in Chernet et al. (2014), Duan et al. (2017), Kay et al. (2006), Raff et al. (2009) or Shamir et al. (2015) is presented in Fig. 4. The possible downscaling techniques available can be divided into dynamical downscaling based on RCMs, statistical downscaling and a

combination of both. Some techniques may be more appropriate than others to simulate precipitation and other extremes (Sarr et al., 2015; Sunyer et al., 2012).

Modelling extreme events remains a challenge, and still more research is needed for analysing and refining the performance of downscaling techniques. Most downscaling techniques are designed to reproduce the mean of the climate signal, which could lead to underestimate the magnitude of the triggering precipitations, although some studies can be found that handle the projection of extreme events (Arnbjerg-Nielsen et al., 2013; Dobler et al., 2013; Pereira-Cardenal et al., 2014; Sarr et al., 2015; Willems, 2012). Other limitations have been identified, for instance, in regions with a complex topography; in such cases, statistical downscaling perform more adequately to generate higher-resolution climate change scenarios (Dobler et al., 2013). Moreover, more attention must be paid to test the influence of non-stationarity in extreme events for flood frequency estimation (Kjeldsen et al., 2014), which is a major uncertainty when applying statistical downscaling techniques (Dixon et al., 2016; Lanzante et al., 2018). Traditionally, frequency analyses are based on the assumption of independency and stationarity of extreme events, which can eventually lead to a miscalculation of the resulting flood quantiles (Šraj et al., 2016; Zhang et al., 2015). Alternative approaches that incorporate the effects of non-independence and non-stationarity (for instance, by using time varying distribution parameters (Khaliq et al., 2006)) can improve the accuracy of the processes. Other attempts seek to reduce these uncertainties related to the statistical downscaling making use of stochastic weather generators (Wilks, 2010) which produce synthetic time series of weather data for a location based on the statistical characteristics of observed weather at that location.

Additionally, impact assessment can benefit from deeper investigations. For instance, uncertainties are inherent to both climate and hydrological projections and should be incorporated to the analyses. These may come from the consideration of several climate models or scenarios (Knutti et al., 2010), but also from the techniques used to obtain a specific projection (e.g., the downscaling method chosen), the hydrological model structure or the parameter identifiability (Chaney et al., 2015). In some cases it can be useful to apply several downscaling methods and compare the results (Willems, 2013).

Studies might also consider the effects of time-varying watershed model parameters in extreme flood climate change studies. For instance, glacier retreat is expected to intensify, leading to an alteration of the flow regimes especially in high mountain regions (Huss et al., 2010). Also, potential evapotranspiration is very likely to increase in a warmer climate, therefore changing the soil conditions when flood events happen. These conditions can in turn influence the generation and propagation of flood hydrographs. Moreover, using flood information separately by seasons can be useful in basins or environments strongly influenced by snow precipitation and storage, where changes in melting of winter snow due to a global warming may play a significant role in peak river runoff (Lawrence et al., 2014).

### 3.1.2 Reservoir water levels

The distribution of the water storage in the reservoir, and thus of the pool levels, determines the loads to which the dam is subjected at the moment of arrival of a flood. A dam with a reservoir that is frequently full will be subjected to higher hydrostatic loads than one with larger fluctuations and less likely to be full. This is captured in the curves representing the relation between water pool level and probability of exceedance for two different cases (Fig. 5): the continues curve represents

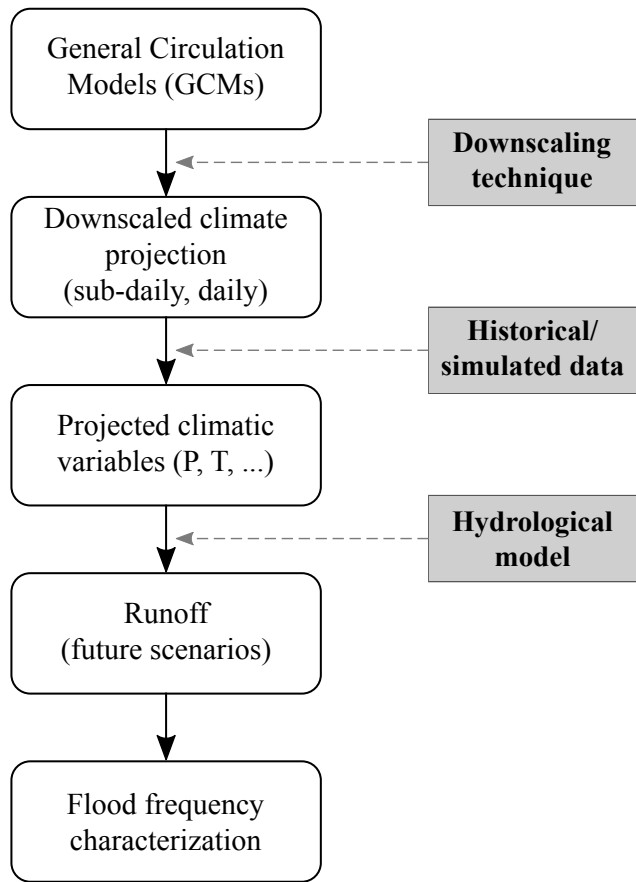

**Figure 4.** Example methodology for the frequency analysis of floods under climate scenarios based on downscaled projections.

the case of Reservoir A which is almost full (level above 540 m a.s.l.) almost 80 % of the time; (b) while the discontinuous cure represents Reservoir B which is half empty more than 70 % of the time. Such distributions depend basically on the inflows, the demands, the reservoir management rules and the water losses (evaporation, infiltration, etc.), and can be obtained either by using the register of historic pool levels or through the simulation of the system of water resources management.

5    Under climate change, surface water availability is expected to fluctuate mainly due to increased precipitation variability (IPCC, 2014) and potential evapotranspiration associated with global warming (Kingston et al., 2009; Seneviratne et al., 2010). However, other factors such as decreased snow and ice storage (Huss, 2011) may have a significant influence. Changes in agricultural land uses, which accounts for about 90 % of global water consumption, are also expected to impact freshwater systems, affecting both the hydrological processes given in the catchment and the water irrigation needs. Moreover, water

10   demand and allocation are strongly driven by demographic, socioeconomic, and technological changes, such as population growth, changes in land use or the adaptation of the reservoirs' exploitation strategies.

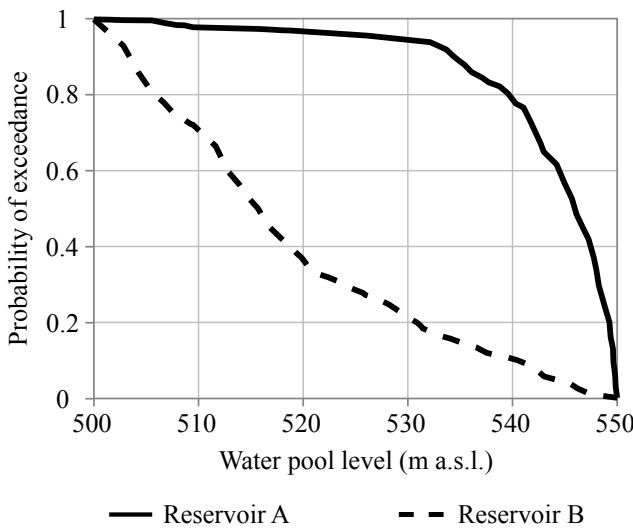

**Figure 5.** Examples of relation between water pool level and probability of exceedance.

The combination of these factors is likely to alter the balance between water availability and supply, and thus will have a direct impact on the water levels in the reservoir. This impact does not only refer to the quantity, but also to the temporal distribution of the water stored, which has a key impact in the dam safety as stated before.

When assessing the effects of climate change on the distribution of the reservoir water levels, analyses must rely on the simulation of the system of water resources management. This allows to reproduce the water balance in the reservoir under specific management rules and for future conditions. Firstly, the inflows are assessed, preferably using long updated climatic series obtained from specific climate models as inputs to a hydrological model. This in turn models the basin behaviour and provides the inflow discharges at the reservoir. These results can be then coupled with the modelling of the system of water resources that computes the allocation and use of the water based on the reservoir's exploitation rules. For complex systems (e.g., the joint operation of several reservoirs), this can be done using simulation tools such as HEC-ResSim (Klipsch and Hurst, 2007) or AquaToolDMA (Andreu et al., 1996). The results of such models are the projected water storage evolution that can be transform in reservoir level series from which the previous pool levels curve can be obtained.

Here too the uncertainties inherent to climate and hydrological projections should be incorporated in the analysis. In this case, the conditions assumed for the water resources' exploitation modelling are also subjected to an uncertainty analysis. Additionally, non-climatic drivers affecting any component of the water balance computation (e.g., changes in land use, adaptation of reservoir's exploitation rules, etc.) can be significant and thus should be included in the analysis. However, the amount of information and work required, and the multiple determining factors involved can turn this procedure impracticable and must be envisaged only when the complexity of the system and the availability of data and time allow it.

### 3.1.3 Gate performance

Spillways and outlet works play a fundamental role in dam safety. They must ensure a certain discharge when required by the arrival of a certain flood. It is therefore important to assess any potential effect that could boost the failure of their regulating gates. Among the different causes that can induce a gates failure, it is worth mentioning (Lewin et al., 2003; SPANCOLD, 2012): human failure, lack of access to the manoeuvre chamber, mechanical failure of the gate or of the civil works, electrical failure, blockage of the outlet works or spillway, or failure in the software controlling the gate or the valve.

An important aspect for their proper working is the good condition of the gates. Severe deficiencies and deterioration could render the outlet works or spillway useless. More intense rainfalls may lead to more soil erosion (Yang et al., 2003) which can be further fostered by changes in land use. Then, an increase in the sediment content of water can worsen the abrasion and erosion processes on the gates, their mechanical equipment or the spillways (British Columbia et al., 1998) thus compromising their reliability. Besides, if the water carries more and bigger suspended material (including trees, branches or debris) this could lead to a blockage of one of the gates, thus reducing the discharge capacity (Paxson et al., 2016). Changes in temperature can also affect the correct manoeuvring of the gates. Hotter or colder conditions, or even greater fluctuations in temperature, can expose gates' mechanisms to additional stresses and/or deformations. This could eventually lead to blockages or malfunctioning of the gates.

The assessment of such impacts on the gates' reliability can be performed using the qualitative description of the gate system's condition. These descriptors are based on standard cases used in dam risk analysis and shown in Table 1, without being necessary to resort to detailed studies such as fault trees (Escuder-Bueno and González-Pérez, 2014). The quantitative individual reliability of the gate (i.e. the probability that it behaves properly) is related to a qualitative description of the condition of the gate system. By estimating the importance of new climatic conditions and stressors such as those mentioned above, one can assess if the gate's state must be updated and thus modify its reliability accordingly.

For more detailed studies, a deeper analysis of the causes and of the assigned failure probabilities is required. The use of fault trees (not to be mistaken for event trees) is a good option to study them in detail (SPANCOLD, 2012; Stamatelatos et al., 2002). Such tools include all the possible manners in which a gate can fail and disaggregate all the failure probabilities, however small they are (Lewin et al., 2003).

### 3.1.4 Flood routing strategy

This component defines how the dam-reservoir system should respond when confronted to a hydrological event. A correct operation of the reservoir allows to maintain adequate safety levels. Such safety levels will also depend on the characteristics of the dam-reservoir system: for some reservoirs the sufficient storage capacity to absorb the inflowing volumes will be determinant, while for others an adequate capacity of releasing peak inflows may be the dominant factor.

Indeed, the routing of the incoming floods reduces the loads to which the dam is subjected. The capability of the dam to perform such routing depends on the state of the outlet works needed to release the discharges as well as on adapted gates operation rules. Potential effects of climate change on these aspects should be checked.

**Table 1.** Standard individual reliabilities of the spillway gates.

| Case | Reliability |
| --- | --- |
| Non-gated | 100 % |
| New / Very well maintained | 95 % |
| Well maintained, some minor problem | 85 % |
| Some problem | 75 % |
| Unreliable | 50 % |
| Unreliable | 0 % |

The operation procedures of the regulated gates establish the desired outflow discharge at any given moment. These procedures will usually be defined depending on a variety of factors, such as the reservoir's water level and its evolution, the inflow discharge, time, etc. Under changes in climate conditions, flood routing strategies are likely to adapt. For instance, the increase of transported sediments driven by soil erosion will accelerate their accumulation within the reservoir, thus impairing the reservoir operation and decreasing its routing capacity and even posing safety hazards to the dam infrastructure (Kondolf et al., 2014; USBR, 2006). Also, changes in heavy rainfall patterns may induce variations in the flood hydrographs concentration time, thus reducing the response capacity. This may compel to re-evaluate operation criteria, especially when relying on methods based on the remaining routing volume such as the Volumetric Evaluation Method (Girón, 1988).

Changes in the reservoir's operation criteria should be analysed under deep analyses that rely on the possible evolution of these criteria attending to climatic and non-climatic drivers. When comparing present and future risk, it is recommended to adapt current operation rules. First, the drivers affecting the definition of the operation rules must be identified. Then, under the consideration of the climate change scenarios adopted, the analysis of the influence on such drivers is performed. Finally, the operation criteria are re-evaluated accordingly. Given the important uncertainty involved in the process, this must be treated carefully to avoid inefficiencies in the analyses; only the most relevant and clear aspects of the problem should be addressed.

## 3.2 System response

### 3.2.1 Failure modes

Failure modes represent the possible ways in which the dam may fail: overtopping, pipping, sliding, etc. Their definition is a key process in risk analysis (FEMA, 2015) since if a relevant failure mode is not included in the analysis, this might lead to an important underestimation of the calculated risk. Different guidelines and tools (FERC, 2005; García-Kabbabe et al., 2010) provide guidance for the identification, description and structuring of new failure modes whenever necessary.

The vulnerability of the dam infrastructure to failure can be somehow affected by climate change. As the conditions of the dam-reservoir system deteriorate or as the climate factors worsen, an update of the failure modes considered may be required. In particular, new failure modes are susceptible to arise or previous ones to become obsolete. For instance, in the context of

geological hazards, studies have confirmed the influence of climate change on slope stability (Damiano and Mercogliano, 2013; Dehn et al., 2000). A slope failure event nearby a dam site could eventually entail a part of the terrain falling into the reservoir or impacting on the dam, which could trigger an overtopping of the dam and eventually a dam failure.

A similar hazard arises in glacial and periglacial environments where the increasing temperatures will likely cause a decrease
in thickness and area of glaciers and progressive permafrost degradation. This thermal perturbation would entail stress redistributions and fast modification of the mechanical conditions at depth (Schneider et al., 2011), which could lead to rock-ice avalanches or Glacial Lake Outburst Floods (GLOFs) entering the reservoir (Evans and Delaney, 2015; Huggel et al., 2008; Stoffel and Huggel, 2012).

### 3.2.2 Probability of failure

Whether new failure modes are taken into account or not, the conditional probability of failure of the dam may also vary under new climatic conditions. To assess such probability and how it is impacted, one can disaggregate each failure mode into its failure mechanisms and then assess the probability of each of them by using different tools (e.g., reliability tools or expert judgment). The objective is to study whether, subject to the same loads, the dam is responding differently under different conditions.

The potential casuistry of climate change effects is large but, for simplicity's sake, in this study only the typical failure modes are examined: overtopping, sliding and internal erosion (piping). For instance, the structural behaviour of concrete dams, and especially arch dams, is directly influenced by temperature (Malm, 2016) and solar exposure (FERC, 1999). Under future climate change, average temperature is expected to increase in all climate scenarios, and may have greater fluctuations during certain periods and reach more frequent extreme values (IPCC, 2013). Moreover, the potential variation of the water storage in
the reservoir (cf. Sect. 3.1.2) can increase the exposition of the body of the dam to sun radiation (both in duration and surface), increasing the temperature difference and causing temperature peaks in the surface of the concrete. These factors can eventually expose the dam to additional mechanical stresses due to the temperature variations, thus turning it more fragile to hydrostatic loads. In these cases, conventional stability analyses may be not sufficient to assess whether the failure probabilities related to dam sliding are influenced by increasing temperatures and solar radiation and then should be adapted. It could thus be of help
performing mechanical and structural analyses, for instance using numerical tools such as finite element or finite difference methods. Similar studies can be applied in case other failure modes (e.g., overtopping or internal erosion) are found influenced by climate change.

In some cases, drier soil conditions are expected due to increasing temperatures and precipitation pattern's variations. Moreover, as stated above, water pool levels may significantly vary and leave the dam at lower levels during long periods. This
could reduce the soil moisture, thus changing the vulnerability of embankment dams to processes such as internal erosion. Indeed, moisture content (and even soil temperature) plays a key role on the internal erosion characteristics (Briaud, 2008). The decrease of water content decreases the critical shear stress and increases the coefficient of piping erosion, thus worsening the soil resistance against erosion (Wan and Fell, 2004). Besides, in dams with vegetated downstream faces, the loss of plants

due to the change in soil moisture may on the one hand leave more or less deep holes that could turn the soil more susceptible to internal erosion processes, and on the other hand present less resistance to surface flow in case of an overtopping event.

Whenever necessary, the assignation of failure probabilities should be complemented with expert consultancy and participatory workshops where results from the models serve as relevant support for the understanding of the problem. More information about probability elicitation through expert judgment can be consulted in different guidelines (ANCOLD, 2003; Ayyub, 2001; SPANCOLD, 2012).

## 3.3 Consequences

Damage produced by a dam failure or an abnormal discharge release is in general very important, causing serious socio-economic consequences. Their analysis is based on two parts: estimation of the outflow hydrographs and their routing downstream, and calculation of the consequences.

### 3.3.1 Outflow hydrographs

An important aspect in the definition of the consequences is the routing of the non-failure and the failure outflow hydrographs. The first one results from the spills released by the outlet works and spillways during the flood routing; the latter one is due to the dam failure. Even if the outflow hydrograph originated by the dam break may be many times greater than in the non-failure case, the impact of climate change is considered analogous and can be analysed jointly.

The study of downstream hydrographs can be split in two stages: estimation of the reservoir outflow hydrograph (through the dam breach or through the outlet works), and routing of the resulting hydrograph throughout the downstream inundation area.

On the one hand, the first stage can be characterized using curves that generally relate the maximum water level reached in the reservoir and the peak discharge (Fig. 6 (a)). These relationships may include other variables depending on the specificities of each case: duration of the hydrograph, speed of the flood wave, etc. According to the hydraulic behaviour of the outflow hydrographs, there are no funded evidences that suggest relevant impacts of climate conditions on this aspect.

On the other hand, these outflow hydrographs are routed to estimate the resulting inundation maps downstream. This information is used to calculate the consequences curve in case of peak discharge (Fig. 6 (b)).

Land use changes can affect substantially the progression of the downstream inundation wave depending on the type of surface (e.g., urbanized or vegetation), its slope, etc. (Bornschein and Pohl, 2018; De Roo et al., 2001). Some studies have applied different techniques and models to forecast future land uses, which can be found in the literature (cf. Sect. 3.3.2). Furthermore, climatic factors such as temperature, precipitation or carbon dioxide concentration are likely to influence plant growth (Morison and Morecroft, 2007; Peñuelas et al., 2004) with a high variability in time and space. This will not only induce a transformation of soil cover (upstream and downstream the dam) but will also affect the amount of sediment contained in the reservoir at the time of the flow release (Braud et al., 2001; Liu et al., 2014). In addition, some studies demonstrate that vegetation cover (Anderson et al., 2006; Järvelä, 2002) and incoming flood sediment concentration (Carrivick, 2010) may influence the propagation of downstream hydrographs. The flow resistance of vegetation increases the roughness of floodplains and then

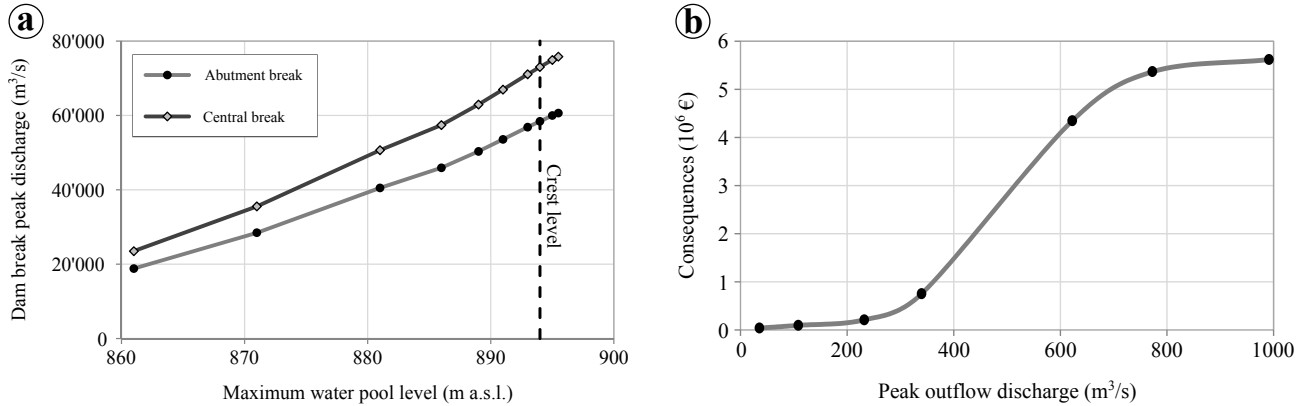

**Figure 6.** (a) Example of relation between the maximum water level attained in the reservoir and failure peak discharge, depending on the failure mode considered (abutment or central break). (b) Example of discharge-consequences curve.

attenuates wave celerity and dispersion of hydrographs, while suspended sediment concentration changes fluid viscosity thus affecting the acceleration and/or deceleration of the flow.

These two main factors —surface roughness and water viscosity— are typically used in floodplain models for the definition of inundation maps (Bladé et al., 2014; DHI, 2014; USACE, 2011a). By updating these inundation models with the projected values of the factors it is possible to analyse their effect on the outflow hydrographs.

### 3.3.2  Socio-economic consequences

Once the downstream hydrographs are defined, it is possible to assess their consequences. A distinction can be made between the direct consequences – those created directly by the impact of the inundation wave – , and the indirect consequences – induced by the direct impacts and which may occur outside the inundation event – (Merz et al., 2004).

Direct consequences:

On the one hand, the calculation of direct consequences due to inundations relies on two factors: exposure, which reflects the presence of people, livelihoods, infrastructure or assets in an at-risk area; and vulnerability, which refers to their propensity to be adversely affected (Cardona et al., 2012; IPCC, 2012a). For the assessment of the impact on direct consequences, changes in exposure and vulnerability are analysed.

According to long-term disaster records, some studies have revealed an increase in the losses due to extreme weather events (Mechler and Kundzewicz, 2010; Peduzzi et al., 2009; Swiss Re, 2016; UNISDR, 2009). The long-term trends in these losses are attributed to the increasing exposure of people and economic assets in at-risk areas due to population and economic growth (Bouwer, 2011; Changnon et al., 2000; Miller et al., 2008; Pielke Jr. et al., 2005) rather than to climatic drivers (Choi and Fischer, 2003; Crompton and McAneney, 2008; Neumayer and Barthel, 2011). This can be extrapolated to inundations (Barredo, 2009; Hilker et al., 2009; Pielke Jr. and Downton, 2000) and hence to the dam risk framework. Indeed, potential increases in

the socio-economic losses are directly influenced by the enhancing presence of people and economic assets in at-risk areas due to population and economic growth (Handmer et al., 2012).

It is also expected that vulnerability facing flooding events will be affected, especially when referring to population vulnerability in poor or underdeveloped environments. Indeed, the capacity to anticipate and respond to inundation risk depends on the existence of public education on risk, warning systems, or coordination between emergency agencies and authorities (Escuder-Bueno et al., 2012). In a changing world, these capacities may vary with the socio-economic development. For example, a potential reduction of economic support for population training or for the maintenance of warning systems may entail a reduced capacity of response, thus increasing its vulnerability.

The assessment of the change in the exposure and vulnerability of the at-risk population and assets due to non-climatic drivers depends on the population and economic growth and should be based on socioeconomic development, urbanization, and infrastructure construction information. Few works have studied jointly both factors when assessing losses from climate change (Hall, 2003; Pielke Jr., 2007; Schmidt et al., 2009).

Regarding population growth, a simple approach could be considering the past demographic evolution at the affected areas and extrapolating it to future scenarios. Another option is using existing projections at regional or national scale such as those available at the online publication Our World in Data (2018), extracted from the UN database (United Nations, 2017). If no specific data are available and due to the complexity of proceeding otherwise, it can be considered that the same current assets and services at risk remain in the future. Only the update of their economic value (cost) is to be applied. Bouwer et al. (2010) propose the application of a factor reflecting the estimate of the increase in value of the at-risk assets based on the index for annual change in gross domestic product (GDP). Results of long term forecasts for the GDP for different countries (up to 2060) can be found in OECD (2018), which are based on an assessment of the economic climate in individual countries and the world economy.

More detailed projections (population, land use and value of assets) can be achieved based on quantitative indicators of societal and economical changes and on the application of specific land use and population growth models. For instance, Maaskant et al. (2009) use projections and spatial distribution of population extracted from a land use model (Schotten et al., 2001) under a high economic growth scenario. Although this scenario was specifically developed for the Netherlands, useful indications can be obtained from other work or guides for the definition and application of socio-economic scenarios (Riahi et al., 2017; UK Climate Impacts Programme, 2000). These practices are often complex and seldom applied (Feyen et al., 2008). Indeed, results of the application of such scenarios are highly dependent on the chosen scenario(s) and must include the corresponding uncertainty. Moreover, land use and economic models can be based on individual behaviour and microeconomic trends that are difficult to capture.

Regarding changes in the population vulnerability, there are different methodologies to assess the inundation severity levels according to the socio-economic context. Escuder-Bueno et al. (2011) propose a classification to assess potential loss of life in urban areas in case of river flooding depending on several factors. Once a socio-economic scenario has been chosen, it is possible (although not always easy) to study how these factors will evolve and then update the vulnerability accordingly.

Indirect consequences:

On the other hand, climate change may have an influence in the indirect consequences. In particular, services and products related to water are of special importance in the context of climate change. Indeed, the value of water allocated to irrigation or hydropower production is likely to vary due to the expected alteration of the distribution, volume and timing of water resources in the future (Fischer et al., 2007; Rodríguez Díaz et al., 2007; Solaun and Cerdá, 2017; U.S. Department of Energy, 2013). Dams are a key component when assessing socio-economic scenarios and their importance may even increase under future climatic conditions (more frequent droughts and extreme events, for instance). Thus, in case of dam failure or serious malfunctioning, the absence of the structure would indeed induce some consequences caused by the fact of being unable to manage part of the water volume.

The assessment of how climate change may impact the indirect consequences is often very complex given the number of components involved and their interrelations. When a deep analysis may be impracticable, indirect costs can be estimated as a fix percentage of direct cost (James and Lee, 1970; SPANCOLD, 2012). This fix percentage could be simply applied to the direct costs that must be re-evaluated under the new climate change scenarios. When the application of a fixed percentage may lead to important errors (e.g., in the case of an airport, for which the indirect costs involved by the interruption of the aerial traffic are much more important than the direct ones), a more detailed work is advised.

Deeper analyses require complex modelling of the economic system to assess how it would be affected by the impact of climate change. First, if it is not yet carried out, an identification of the potentially affected services and economic activities is required (e.g., electric or telecommunications supply, industrial production). Then, specific models are to be used to assess the indirect costs induced by the interruption of these services and/or activities due to a dam failure or disruption event. Different methodologies, such as the Input-Output or the Computable General Equilibrium analyses (U.S. Department of Homeland Security, 2011), can be applied to study the variations of the economic flows after the flood. An analysis of how each component used in these models is susceptible to change in the future must be done.

In order to simplify the work, one can study only the impact in the most relevant activities affected (e.g., services and products related to water such as irrigation or hydropower production). Different works and methodologies have been developed to analyse how climate change may affect the resulting damage on the water resources systems (Hutton et al., 2007; Kazem et al., 2016; Quiroga et al., 2011).

## 3.4 Summary

A succinct summary of the main impacts identified for each dam risk component is presented in Table 2 along with some recommended techniques and methods for their assessment.

## 4 Conclusions

This work presents an interdisciplinary review of the state-of-the-art research on projected climate change impacts on dam safety attending to both climatic and non-climatic drivers. The structure followed for such review is based on the risk analysis approach where all the variables concerning dam safety – from the hydrological loads to the consequences of failure – and their

**Table 2.** Summary of climate change impacts on the different dam safety components and suggested methods for their assessment.

| Risk component | Climate change impacts | Assessment methods |
|---|---|---|
| Flood | Variations in local floods are expected due to changes in:<br>– Heavy rainfall patterns.<br>– Snow cover and snowmelt processes.<br>– Vegetation or soil moisture. | – Direct application of previous analyses.<br>– Combination of climate projections, downscaling and hydro-meteorological modelling (Fig. 4).<br>– Uncertainties inherent to climate and hydrological projections and changes in watershed model. |
| Reservoir water levels | Fluctuations of water storage due to:<br>– Precipitation variability, potential evapotranspiration or decreased snow and ice storage.<br>– Changes and adaptations in land use and water demand. | – Combination of climate projections, downscaling techniques and simulation of the system of water resources management.<br>– Importance of non-climatic drivers (e.g., changes in land use, adaptation of reservoir's exploitation rules). |
| Gate performance | – Abrasion processes due to increase in the sediment content of the water.<br>– Blockage of the gates due to suspended material.<br>– Changes in temperature causing stresses and deformations. | – Qualitative assessment of the impacts of new climatic conditions and stressors (Table 1).<br>– Use of fault trees. |
| Flood routing strategy | Operation rules are likely to adapt under certain climate conditions (e.g., changes in heavy rainfalls inducing variations in the flood hydrographs concentration time). | Re-evaluation of the flood routing criteria. |
| Failure modes | New failures modes are susceptible to arise, in particular in the context of glacier melt and slope stability or GLOFs occurrence directly impacting the dam structure. | Guidelines and tools to identify, describe and structure new failure modes or remove obsolete ones. |
| Probability of failure | – Temperature fluctuations may induce additional mechanical stresses in concrete dams.<br>– Drier soils and water level fluctuations can increase processes such as internal erosion in embankment dams. | Probability elicitation through expert judgment in different guidelines. |
| Outflow hydrographs | The outflow hydrograph routing is affected by:<br>– Surface roughness of the surface.<br>– Water viscosity related to flood sediment concentration. | Use of inundation models to assess the sensitivity of the outflow hydrographs to these factors. |
| Socio-economic consequences | Direct consequences:<br>– Exposure changes due to population growth.<br>– Update of the assets' economic value.<br>Indirect consequences:<br>– The value of water for irrigation or hydropower production is likely to vary, which implies changes in the cost of interruption of services and/or activities. | Direct consequences:<br>– Application of demographic projections.<br>– Detailed land use and population growth models based on socio-economic scenarios.<br>– Assessment of flood severity levels according to the socio-economic context.<br>Indirect consequences:<br>– Estimation as a fix percentage of direct costs.<br>– Complex modelling of the economic system and assessment of costs induced by the interruption of services and/or activities. |

interdependencies are included in a comprehensive way. The extent of the analysis to be performed should depend on the detail level chosen. Paired with the impacts identified, the paper also presents the useful techniques for their direct application to provide information for dam owners and dam safety practitioners in their decision-making process. Although the information collected in this work is mainly based on existing works, there is still some novelty or innovation in its processing since usually the global effects of climate change on dam risk are studied separately. The authors introduce a more comprehensive and structured approach to take them into account, which can be used to apply this same risk analysis to other critical infrastructures.

The purpose of this review is to serve as a dam safety management supporting tool to assess the vulnerability of the dam to climate change, i.e. the additional risk imposed by climate change effects, and to define adaptation strategies for new climate scenarios under an evolutive dam risk management framework (Fig. 7). Under this approach, dam risk models must be updated following to the effects of climate change on each of the risk components, which will later help defining the adaptation strategies to be followed. As climate projections evolve with new scenarios of models, the process must be replicated iteratively.

With this information, long-term investments can be planned more efficiently. Indeed, the application of such tool may prevent investing in measures that would no longer be necessary in the future, or missing some measures that could reduce the future risk. As such, it is addressed to dam owners and dam safety practitioners, but also to the research community that can help improving it and filling the gaps that still remain in some aspects of the risk assessment.

The present work is based on available data sources and information at current levels of knowledge. However, this filed of research is highly dynamic and advances in science and techniques for the assessment of these climate change effects are expected over time. Therefore, climate change impacts can then be iteratively actualized along with the forthcoming innovations and advances in science and techniques for the assessment of these effects. In particular, climate modellers as well as dam engineers face significant uncertainties when proposing and assessing climate scenarios and their impact on the different components involved in dam safety. The assignation of probabilities to uncertain future conditions and scenarios remains a major challenge and thus the management of dam safety based on climate change impacts must take into account these limitations.

*Author contributions.* Javier Fluixá-Sanmartín prepared the manuscript with contributions from all co-authors.

*Competing interests.* The authors declare that they have no conflict of interest.

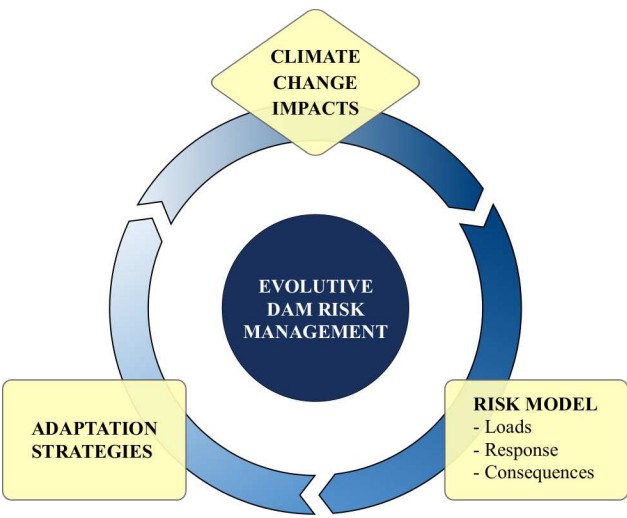

**Figure 7.** Evolutive dam risk management driven by climate change impacts on risk components, including adaptation strategies.

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
