# Peer review of "Review article: climate change impacts on dam safety"

_Natural Hazards and Earth System Sciences, 2018_

## Referee Comment (RC1) · Anonymous Referee #1 · 21 Jun 2018

This is a comprehensive review paper on climate change impacts on dam safely. The authors review a large number of relevant papers and technical documents published on this subject in the last years. The main scientific value of the work is certainly that information from many different sources has been collected, organized and presented in one single study. The authors state their objectives at the end of the introduction: a multidisciplinary and structured review of the impacts of climate change on the relevant dam safety components. With these goal in mind they present their analysis following the traditional risk analysis approach. Their emphasis is on dam safety components: loads, system response and consequences. In my opinion, they clearly fulfil their objectives. The state of the art in the different dam safety components is extensively reviewed. While some fields, like flood hydrology, have been frequently studied, other

fields, like gate reliability or socio-economic consequences, have received less attention. Overall, the paper is correctly organized and well written. In conclusion, the topic is interesting, the analysis is sound, the review is useful and the presentation is clear. In my opinion, it can be published in NHESS.

I only have a couple minor suggestions for the consideration of the authors. I understand that the paper is already long and with such a broad topic the authors have obviously to choose the material that is included in the presentation, but I particularly missed two discussions that I believe are of interest for the general reader.

While introducing floods, the authors state that they are usually characterized through a relation between their peak discharge Q and volume V, and the associated annual exceedance probability. In my opinion, this sentence needs some clarification. In order to define an exceedance probability for a hydrograph (combination of Q and V), a way of comparing two hydrographs is required. In the context of dam safety analysis, this comparison should be made through the maximum water level that the hydrograph attains while it is routed through the reservoir. For some reservoirs, Q is the most relevant variable, while for other V may be the dominant factor. I suggest that the authors add a brief explanation to clarify this issue.

I also missed a discussion of non-stationary flood frequency analysis. For some dams the hydrological load is mostly estimated through flood frequency analysis. Many studies have been carried out to account for climate change through non-stationary models and a brief discussion of this work would be adequate for a review paper.

One more detail: Section 3.14 is titled "Flood routing". I think "Flood management strategy" may be more appropriate, since the emphasis is not on how you compute the flood wave propagation through the reservoir, but on how you decide how to manage the flood wave to minimize dam risk and downstream damages.

---

## Referee Comment (RC2) · Anonymous Referee #2 · 23 Jun 2018

The review is clearly structured and easy to read. I also think that most of the text is substantiated with good references and makes sense. Novelty cannot be demanded from a review, but I still miss some recommendations coming out of the review: What is the main findings of collecting this information, either for practice to be used for design or operation or for the research community to explore further because of lack of information on specific or cross-cutting topics. As it is I am in doubt what the purpose of the paper is.

Throughout the paper I am in doubt whether the safety assessment is related to up-stream or downstream of the dam itself or both. The discussion on the hydrological performance seems to focus on upstream loading, but typically dam safety is also concerned with downstream consequences of failures. Please clarify the scope of the

paper, also in terms of physical boundaries. When considering T=100000 downstream consequences can be catastrophic and often much higher than upstream.

The risk definition in Eq 1 [P3L3] is unusual and not very accurate. 'Events under study' imply a finite number of events and hence summation rather than integration. The cited ref does not give an explation and I cannot identify the ICOLD (2003) reference, which seems to be the original reference. If you wish to use that definition I suggest you start with a standard formulation as in e.g. Merz et al (2010) and then extend to the definition you wish to apply, including a discussion of Figure 1. Together they can form a precise definition. It would be nice to end the review with how this figure (that only considers stationary input) by means of this review can be converted into a diagram, that considers nonstationarity. As a community we miss a figure that combines the detailed analysis schemes corresponding to Fig01 with the soft figures from IPCC (e.g. http://www.ipcc.ch/report/graphics/images/Special%20Reports/SREX/Chapter%2001/Fig1-1.jpg from IPCC (2012)).

I am somewhat surprised that the review does not specifically cover the impacts in relation to the frequency domain. Climate change (and other drivers) impact both average and extreme values. Intuitively the first will impact operation, including the value of having a dam, while the second must be most important for safety. I sometimes see the paper only focussing on extremes (e.g. in the hydrology section) but other times also average impacts (e.g. socio-economic section).

There is an interesting mentioning on other key drivers of change such as population increase, economic development etc [P4L10] but the paper does not come back to discussing this except for a very broad discussion on socio-economic consequences. During this discussion I cannot see if it is the presence of the dam that drives changes in the socio-economic conditions or vice versa. Is there a difference between deciding on building of a dam and considering safety in relation to an existing dam?

More detailed comments: [Fig01] Diagram starts with a loading of Flood. Is that up-

stream or downstream of the dam? A Consequence (Hydrograph (no failure)) leads to Failure modes, which seems counterintuitive. Figure needs more explanation.

[4L15] You use the term flooding both for a high loading of the dam and for the consequences downstream. The illustrations in Fig02 seems to be unaffected by dam management, but is discussed as if it is downstream?

[P6L15] There is quite abundant literature on extracting correct projections of extremes. For single sites methods developed by e.g. Kilsby and Willems are predominant (each has many refs), while for larger regions spatially distributed methods are employed (e.g. Pereira-Cardenal et al 2014 incl. supplementary information).

[Fig04] This forward-propagation modelling chain deserves a reference.

[Fig05] Subfigure a and b are identical. Suggest to replace legend on yaxis on subfigure a.

References: IPCC (2012). Managing the Risks of Extreme Events and Disasters to Advance Climate Change Adaptation (SREX). Downloaded from http://www.ipcc.ch/report/srex/

Merz B, Hall J, Disse M, and Schumann A (2010). Fluvial flood risk management in a changing world. NHESS, https://www.nat-hazards-earth-syst-sci.net/10/509/2010/nhess-10-509-2010.pdf

Pereira-Cardenal SJ, Madsen H, Arnbjerg-Nielsen K, Riegels N, Jensen R, Mo B, Wangensteen I, and Bauer-Gottwein P. 2014. Assessing climate change impacts on the Iberian power system using a coupled waterpower model. Climatic Change, 126, 3-4, 351-364. DOI 10.1007/s10584-014-1221-1.

---

## Short Comment (SC1) · 27 Jun 2018

Dear authors, I enjoyed reading about this particular problem from different perspectives. I agree it's important to not only consider hydro-meteorological variables but also an engineering and a socio-economic point of view. The article is comprehensive in terms of different aspects affecting dam safety. However, I think that a more comprehensive literature review on the single aspects could help the manuscript being an even more valuable contribution to the scientific discussion. The following questions, mainly focusing on the system load component of your risk analysis approach, may provide some inspiration.

1) Isn't there a mismatch between the target audience ("dam owners and dam safety

practitioners") and the presented approaches? If the article aims to provide information for the mentioned target audience, I doubt that a methodology as proposed in Figure 4 is very useful for them. There are probably not many dam owners or dam safety practitioners who are capable to conduct GCM runs and statistical or dynamical downscaling procedures. Moreover, many dams are built within a complex topography, where results from global analyses are less reliable, and a proper downscaling of extreme events is even more challenging. Maybe this is a bit beyond the scope, but the question arises who should provide such locally representative scenarios. Science, governmental agencies, dam owners or someone else? On the other hand, if the article aims to contribute to the scientific discussion, I would suggest to elaborate a bit more on the literature review, particularly regarding GCM downscaling procedures for extreme events and regarding the stationarity assumption in FFE.

2) From a system load perspective, figure 4 implies that a downscaling of extreme GCM scenarios (or even long term runs) is the only way to assess climate change impact on dam safety risk. What about other (less costly) approaches like non-stationary FFE, or the use of adapted stochastic weather generators?

3) The probability of dam failure from a hydro-meteorological point of view is not primarily a question of the peak inflow, it's more dependent on the total inflowing volume over a given amount of time. I would therefore suggest to mention the ongoing discussion on bivariate FFE methods as well.

4) I'm not sure whether the climate change impact on land use change and surface roughness is relevant for this topic. First, most of the hydrodynamic models for flood mapping are calibrated using the roughness parameter. A change of such a calibrated parameter could cause non-linear changes in the simulated runoff, and might lead to wrong conclusions. Second, I would consider slight changes in roughness as negligible, considering the huge uncertainties that come along with FFE (not shown in Figures 2a and 2b) and other parts of the system load component. More generally spoken: can you say something about the sensitivity of the single components in Figure 1?

---

## Author Comment (AC1) · 17 Jul 2018

These are the Authors' replies to comments from Referee #1, received and published on 21st June, 2018. We use blue color for our replies and black color for Referee's comments.

**RESPONSES**:

Firstly, we want to sincerely thank Referee #1 for the remarks and recommendations which will undoubtedly improve the quality and scope of the paper.

While introducing floods, the authors state that they are usually characterized through a relation between their peak discharge Q and volume V, and the associated annual exceedance probability. In my opinion, this sentence needs some clarification. In order to define an exceedance probability for a hydrograph (combination of Q and V), a way of comparing two hydrographs is required. In the context of dam safety analysis, this comparison should be made through the maximum water level that the hydrograph attains while it is routed through the reservoir. For some reservoirs, Q is the most relevant variable, while for other V may be the dominant factor. I suggest that the authors add a brief explanation to clarify this issue.

When mentioning Q and V (and their relation with AEP), the authors refer to hydrological variables on which frequency analyses can be applied to estimate the occurrence probabilities of such floods. Further variables (such as the maximum water level that the hydrograph attains while it is routed through the reservoir) depend on the magnitude and frequency of these floods, but also on other factors like the gate performance or the previous reservoir level and thus cannot be directly used to compare 2 hydrographs.

For clarity, the authors suggest the following paragraph:

"*In the hydrological scenario, floods are the initiating event (node) that creates the loads to which the dam is subjected. The probabilities of the emerging branches are defined by the frequecy occurrence linked to the inflow hydrographs (Figure 2 (b)), introducing the temporal component to the risk calculation [consequences/**year**]. These are associated with a given return period (T) or its equivalent annual exceedance probability (AEP).*

*Different analyses can be performed to estimate the occurrence probability of these events using deterministic, parametric, probabilistic and stochastic methods (World Meteorological Organization, 2008). Some of them seek relating the magnitude of one or more hydrological variables with T. A widely used approach to characterize this relation is to perform univariate or bivariate frequency analyses of the maximum values of peak discharge ($Q_P$) and/or volume (V) (Figure 2 (a)).*"

Regarding the relative influence of Q and V in dam safety, it is clear that this will depend on the characteristics of the dam-reservoir system (a soon to be published work of the authors characterizes the factors affecting this relation). In the approach followed, this influence is revealed when routing the incoming floods through the reservoir (nodes **Maximum pool level** and **Hydrograph (no failure)**).

I also missed a discussion of non-stationary flood frequency analysis. For some dams the hydrological load is mostly estimated through flood frequency analysis. Many studies have been carried out to account for climate change through non-stationary models and a brief discussion of this work would be adequate for a review paper.

The authors agree with Referee #1 about the convenience of including a special mention of non-stationary flood frequency analysis and citing relevant studies dealing with this subject (e.g., Gilroy and McCuen, 2012; López and Francés, 2013).

One more detail: Section 3.14 is titled "Flood routing". I think "Flood management strategy" may be more appropriate, since the emphasis is not on how you compute the flood wave propagation through the reservoir, but on how you decide how to manage the flood wave to minimize dam risk and downstream damages.

The title "Flood routing" was directly extracted from the document of SPANCOLD (2012) from where the risk modelling approach used in the paper (Figure 1) was taken. Following the title suggested by Referee #1 ("Flood management strategy"), the authors propose to combine both in "Flood routing strategy", which will replace the title of Section 3.1.4 as well as in Figure 1.

**REFERENCES**

Gilroy, K. L. and McCuen, R. H.: A nonstationary flood frequency analysis method to adjust for future climate change and urbanization, Journal of Hydrology, 414–415, 40–48, doi:10.1016/j.jhydrol.2011.10.009, 2012.

López, J. and Francés, F.: Non-stationary flood frequency analysis in continental Spanish rivers, using climate and reservoir indices as external covariates, Hydrology and Earth System Sciences, 17(8), 3189–3203, doi:10.5194/hess-17-3189-2013, 2013.

SPANCOLD: Risk Analysis as Applied to Dam Safety. Technical Guide on Operation of Dams and Reservoirs, Professional Association of Civil Engineers. Spanish National Committe on Large Dams, Madrid. [online] Available from: http://www.spancold.es/Archivos/Monograph_Risk_Analysis.pdf, 2012.

World Meteorological Organization: Guide to hydrological practices, World Meteorological Organization, Geneva., 2008.

---

## Author Comment (AC2) · 17 Jul 2018

These are the Authors' replies to comments from Referee #2, received and published on 23[rd] June, 2018. We use blue color for our replies and black color for Referee's comments.

It should be noted that the comments of the Referee make reference to pages and lines of the production paper. We use the same criteria in this reply.

**RESPONSES**:

Firstly, we want to sincerely thank Referee #2 for the remarks and recommendations which will undoubtedly improve the quality and scope of the paper.

Novelty cannot be demanded from a review, but I still miss some recommendations coming out of the review: What is the main findings of collecting this information, either for practice to be used for design or operation or for the research community to explore further because of lack of information on specific or cross-cutting topics. As it is I am in doubt what the purpose of the paper is.

The purpose of the manuscript is to serve as a dam safety management supporting tool when assessing how climate change may affect dam risk. With this information, long-term investments can be planned more efficiently. Indeed, the application of this tool may prevent investing in measures that would no longer be necessary in the future, or missing some measures that could reduce the future risk. As such, it is addressed to dam owners and dam safety practitioners, but also to the research community that can help improving it and filling the gaps that still remain in some aspects of the risk assessment.

Moreover, although the information collected in this work is mainly based on existing works, there is still some novelty or innovation in its processing since usually the global effects of climate change on dam risk are studied separately. The authors introduce a more comprehensive and structured approach to take them into account, which can be used to apply this same risk analysis to other critical infrastructures.

These remarks will be added to the new version of the manuscript.

Throughout the paper I am in doubt whether the safety assessment is related to upstream or downstream of the dam itself or both. The discussion on the hydrological performance seems to focus on upstream loading, but typically dam safety is also concerned with downstream consequences of failures. Please clarify the scope of the paper, also in terms of physical boundaries. When considering T=100000 downstream consequences can be catastrophic and often much higher than upstream.

Dam risk analyses focus on the downstream consequences of a dam failure but depend also on the upstream conditions or loads. Thus, in terms of physical boundaries, the dam safety assessment can be divided in two parts that correspond to 2 different components of dam risk:

-   When we assess the loads of a dam in the hydrological scenario, including the incoming floods to the reservoir, we refer to upstream of the dam.
-   When we assess the consequences we refer to downstream of the dam.

For clarity, this will be mentioned in the new version of the manuscript.

The comment regarding the consideration of floods of T=100'000 years return period unveils an interesting question. As mentioned by Escuder-Bueno et al. (2012), the presence of dams changes

the downstream risk. **Figure 1** shows how F-D curves[1] are modified when structural measures (such as dams) are implemented: compared to situations without any measures, the F-D curves show a decrease in the downstream consequences for a certain rank of annual probabilities of exceedance, but also an increase in these consequences for lower probabilities (e.g. corresponding to the breakage of the dam) which is consistent with the Referee's comment.

Moreover it is worth noting that in Risk Analysis it is common practice to deal with incremental risk, that is the part of risk exclusively due to the dam failure which is obtained by subtracting from the consequences of the dam failure the ones that would have happened anyway (even if the dam had not failed). In case of a T=100'000 flood, the consequences downstream must be contrasted with the consequences of the dam break.

[Figure]

**Figure 1: Effect of structural and non-structural measures on the F–D curve (from Escuder-Bueno et al., 2012).**

The risk definition in Eq 1 [P3L3] is unusual and not very accurate. 'Events under study' imply a finite number of events and hence summation rather than integration. The cited ref does not give an explation and I cannot identify the ICOLD (2003) reference, which seems to be the original reference. If you wish to use that definition I suggest you start with a standard formulation as in e.g. Merz et al (2010) and then extend to the definition you wish to apply, including a discussion of Figure 1. Together they can form a precise definition. It would be nice to end the review with how this figure (that only considers stationary input) by means of this review can be converted into a diagram, that considers nonstationarity. As a community we miss a figure that combines the detailed analysis schemes corresponding to Fig01 with the soft figures from IPCC (e.g. http://www.ipcc.ch/report/graphics/images/Special%20Reports/SREX/Chapter%2001/Fig1-1.jpg from IPCC (2012)).

The risk definition concept in Eq 1 [P3L3] was set in 1981 in the classical paper by Kaplan and Garrick (1981). This definition has been used extensively across different sectors in the industry and has been discussed in Aven (2012). In the context of dam safety, it has been frequently used and can be found,
* * *
[1] An F-D curve illustrates the cumulative annual exceedance probability of the expected economic damages.

for instance, in Serrano-Lombillo et al. (2011). Nevertheless, reference to Merz et al. (2010) will be included in the manuscript.

Regarding the mutable aspect of Figure 1, as suggested by Referee #2 the authors will prepare a new diagram that will be included in the manuscript.

I am somewhat surprised that the review does not specifically cover the impacts in relation to the frequency domain. Climate change (and other drivers) impact both average and extreme values. Intuitively the first will impact operation, including the value of having a dam, while the second must be most important for safety. I sometimes see the paper only focusing on extremes (e.g. in the hydrology section) but other times also average impacts (e.g. socio-economic section).

The approach followed in the manuscript allows to integrate all the different types of aspects (from normal to extreme events) of dam safety that may be affected by climate change. On one hand, when performing a risk analysis focused on a hydrological scenario as proposed in the manuscript, the main loads are floods. In this case the analysis of climate change impacts on extreme hydrological conditions is required. On the other hand, other components such as the previous pool level or the population exposure downstream represent the normal state of these factors and have to be analyzed considering them as averaged values. That is why at certain point the paper focuses on extremes and other times in average conditions.

For clarity, this should be mentioned more explicitly in the new version of the manuscript.

There is an interesting mentioning on other key drivers of change such as population increase, economic development etc [P4L10] but the paper does not come back to discussing this except for a very broad discussion on socio-economic consequences. During this discussion I cannot see if it is the presence of the dam that drives changes in the socio-economic conditions or vice versa. Is there a difference between deciding on building of a dam and considering safety in relation to an existing dam?

The presence of the dam certainly represents an important driver to eventual socio-economic changes. Dams are indeed a key component when applying socio-economic scenarios and their importance may even increase under future climatic conditions (more frequent droughts and extreme events, for instance).

When assessing the future safety level of a dam, we could consider that there are no differences between applying the methodology to an existing dam or to a future non-existing dam. However, when assessing the safety of an existing dam, it is likely that the population in at-risk areas has already followed several safety procedure trainings which would entail a better prepared population facing inundation events thus changing their vulnerability in comparison with a non-trained population.

[Fig01] Diagram starts with a loading of Flood. Is that up- stream or downstream of the dam? A Consequence (Hydrograph (no failure)) leads to Failure modes, which seems counterintuitive. Figure needs more explanation.

The **Flood** node of Figure 1 refers to an upstream flow. Downstream flows are referred as **Hydrograph (failure and no failure)**.

The arrows in Figure 1 represent the information flow from the start until the end of the diagram. In some cases, the information transmitted is used on the connected node (for instance, some information transmitted by the node **Gate performance** is used to define the **Maximum pool level** node). However, in other cases the information is only used a little further in the diagram (for

instance, the information contained in the node **Flood** is not directly used by the node **Previous pool level**, and yet they are both connected by an arrow).

It is true that this case is particularly tricky when connecting **Hydrograph (no failure)** and **Failure modes** since it may lead to a misunderstanding. That is why the authors suggest using this diagram instead, which contains the same information:

[Figure]

[Figure]

**Legend for nodes:**

| Loads | System Response | Consequences |

[4L15] You use the term flooding both for a high loading of the dam and for the consequences downstream. The illustrations in Fig02 seems to be unaffected by dam management, but is discussed as if it is downstream?

The use of the term flood in both cases can indeed lead to a misunderstanding of whether we refer to upstream or downstream. Thus, the authors suggest using the term flood exclusively for the upstream flow into the reservoir and outflow hydrograph or inundation when referring to downstream the dam.

Figure 2 corresponds to the floods as a load to the dam (upstream).

[P6L15] There is quite abundant literature on extracting correct projections of extremes. For single sites methods developed by e.g. Kilsby and Willems are predominant (each has many refs), while for larger regions spatially distributed methods are employed (e.g. Pereira-Cardenal et al 2014 incl. supplementary information).

The references suggested by Referee #2 cover adequately the scope of the work and will be added to the final version of the manuscript when referring to the projection of extremes.

[Fig04] This forward-propagation modelling chain deserves a reference.

This figure has not been directly extracted from any reference. It has been elaborated by the authors based on the methodologies commonly used in many studies (e.g., Chernet et al., 2014; Duan et al., 2017; Kay et al., 2006; Raff et al., 2009; Shamir et al., 2015) and serves as an example of procedure to perform frequency analysis on floods taking into account the effect of climate change.

The legend of the figure that appears in the manuscript is not accurate and should be changed to "Example of methodology for the frequency analysis of floods under climate scenarios."

[Fig05] Subfigure a and b are identical. Suggest to replace legend on yaxis on subfigure a.

The objective of showing these two figures was to highlight how the distribution of the water storage in the reservoir (and thus the loads to which the dam is subjected at the moment of arrival of a flood) varies depending on the water level regimes.

Figure 5.a) and Figure 5.b) represent the same information (relation between water pool level and probability of exceedance) but for two different cases of dams, and that is why their axes are the same:

- Figure 5.a) shows the case of a reservoir that is almost full (level above 540 m a.s.l.) almost 80 % of the time.
- Figure 5.b) represents the case of a reservoir that is half empty more than 70 % of the time.

For clarity, both curves will be plotted in a unique graphic and an explanatory legend will be included to distinguish one case from the other.

**REFERENCES**

Aven, T.: The risk concept—historical and recent development trends, Reliability Engineering & System Safety, 99, 33–44, doi:10.1016/j.ress.2011.11.006, 2012.

Chernet, H. H., Alfredsen, K. and Midttømme, G. H.: Safety of Hydropower Dams in a Changing Climate, Journal of Hydrologic Engineering, 19(3), 569–582, doi:10.1061/(ASCE)HE.1943-5584.0000836, 2014.

Duan, J. G., Bai, Y., Dominguez, F., Rivera, E. and Meixner, T.: Framework for incorporating climate change on flood magnitude and frequency analysis in the upper Santa Cruz River, Journal of Hydrology, 549, 194–207, doi:10.1016/j.jhydrol.2017.03.042, 2017.

Escuder-Bueno, I., Castillo-Rodríguez, J. T., Zechner, S., Jöbstl, C., Perales-Momparler, S. and Petaccia, G.: A quantitative flood risk analysis methodology for urban areas with integration of social research data, Natural Hazards and Earth System Science, 12(9), 2843–2863, doi:10.5194/nhess-12-2843-2012, 2012.

Kaplan, S. and Garrick, B. J.: On The Quantitative Definition of Risk, Risk Analysis, 1(1), 11–27, doi:10.1111/j.1539-6924.1981.tb01350.x, 1981.

Kay, A. L., Reynard, N. S. and Jones, R. G.: RCM rainfall for UK flood frequency estimation. I. Method and validation, Journal of Hydrology, 318(1–4), 151–162, doi:10.1016/j.jhydrol.2005.06.012, 2006.

Merz, B., Hall, J., Disse, M. and Schumann, A.: Fluvial flood risk management in a changing world, Natural Hazards and Earth System Science, 10(3), 509–527, doi:10.5194/nhess-10-509-2010, 2010.

Raff, D. A., Pruitt, T. and Brekke, L. D.: A framework for assessing flood frequency based on climate projection information, Hydrology and Earth System Sciences, 13(11), 2119–2136, doi:10.5194/hess-13-2119-2009, 2009.

Serrano-Lombillo, A., Escuder-Bueno, I., de Membrillera-Ortuño, M. G. and Altarejos-García, L.: Methodology for the Calculation of Annualized Incremental Risks in Systems of Dams: Risk

Calculation for Systems of Dams, Risk Analysis, 31(6), 1000–1015, doi:10.1111/j.1539-6924.2010.01547.x, 2011.

Shamir, E., Megdal, S. B., Carrillo, C., Castro, C. L., Chang, H.-I., Chief, K., Corkhill, F. E., Eden, S., Georgakakos, K. P., Nelson, K. M. and Prietto, J.: Climate change and water resources management in the Upper Santa Cruz River, Arizona, Journal of Hydrology, 521, 18–33, doi:10.1016/j.jhydrol.2014.11.062, 2015.

---

## Author Comment (AC3) · 17 Jul 2018

**AUTHOR'S RESPONSES TO GUIDO FELDER**

These are the Authors' replies to comments from Guido Felder, received and published on 27[th] June, 2018. We use blue color for our replies and black color for the comments.

**RESPONSES**:

Firstly, we want to sincerely thank Guido Felder for the remarks and recommendations which will undoubtedly improve the quality and scope of the paper.

1) Isn't there a mismatch between the target audience ("dam owners and dam safety practitioners") and the presented approaches? If the article aims to provide information for the mentioned target audience, I doubt that a methodology as proposed in Figure 4 is very useful for them. There are probably not many dam owners or dam safety practitioners who are capable to conduct GCM runs and statistical or dynamical downscaling procedures. Moreover, many dams are built within a complex topography, where results from global analyses are less reliable, and a proper downscaling of extreme events is even more challenging. Maybe this is a bit beyond the scope, but the question arises who should provide such locally representative scenarios. Science, governmental agencies, dam owners or someone else? On the other hand, if the article aims to contribute to the scientific discussion, I would suggest to elaborate a bit more on the literature review, particularly regarding GCM downscaling procedures for extreme events and regarding the stationarity assumption in FFE.

Dam Risk Analysis is a useful tool that must interpellate owners, administrations, regulators, consultants, users, civil protection services and all type of related entities and agents in dam management (SPANCOLD, 2012). The approach presented in the paper summarizes the state of the art of techniques and methodologies available to perform required studies and, although not all dam owners or dam safety practitioners may be capable to conduct some of the proposed studies, other techniques are based on easily applicable works and can be conducted without much effort. The authors would like to avoid pre-established restrictions on what techniques can be applied and who can apply them. In this regard, the authors have participated in more than 40 dam safety analyses in the last 15 years (Escuder-Bueno et al., 2016; Morales-Torres et al., 2016; Serrano-Lombillo et al., 2017; Setrakian-Melgonian et al., 2017) where not only technicians were involved but also dam owners and managers.

On the other hand, the authors agree that a more detailed review on GCM downscaling procedures for extreme events and regarding the stationarity assumption in FFE can improve the scope of the paper and thus new references will be added to the final version of the paper.

2) From a system load perspective, figure 4 implies that a downscaling of extreme GCM scenarios (or even long term runs) is the only way to assess climate change impact on dam safety risk. What about other (less costly) approaches like non-stationary FFE, or the use of adapted stochastic weather generators?

The objective of presenting the approach of Figure 4 was not to state that this was the only way of assessing the effects of climate change on the analysis of floods, but more of an example proposition. Other methods like non-stationary FFE, or the use of adapted stochastic weather generators will also be included in the next version of the paper to widen the scope of the work.

The legend of the figure that appears in the manuscript is not accurate and should be changed to "Example of methodology for the frequency analysis of floods under climate scenarios."

3) The probability of dam failure from a hydro-meteorological point of view is not primarily a question of the peak inflow, it's more dependent on the total inflowing volume over a given amount of time. I would therefore suggest to mention the ongoing discussion on bivariate FFE methods as well.

The relative influence of both factors (peak inflow and total inflowing volume) and/or their combination on failure probability cannot be generalized as it will depend on each case. In this context, a broader assertion could be stated: the probability of dam failure is not **only** a question of the peak inflow **and** the total inflowing volume, but **also** depends on the dam-reservoir system's capacity to absorb such hydrological loads.

If we consider the overtopping failure mode, for instance, its probability of occurrence mainly depends on two connected dam-reservoir characteristics: insufficient flood storage capacity and inadequate spillway capacity (Lee and You, 2013; USBR, 2014). On this matter, a soon to be published work of the authors reveals the relative importance of both the peak discharge and the flood volume when assessing the overtopping probability.

However, the authors consider that the comment is very pertinent in the context of hydrological risk evaluation and thus a special mention of the bivariate flood frequency methodologies will be included in the text.

4) I'm not sure whether the climate change impact on land use change and surface roughness is relevant for this topic. First, most of the hydrodynamic models for flood mapping are calibrated using the roughness parameter. A change of such a calibrated parameter could cause non-linear changes in the simulated runoff, and might lead to wrong conclusions. Second, I would consider slight changes in roughness as negligible, considering the huge uncertainties that come along with FFE (not shown in Figures 2a and 2b) and other parts of the system load component. More generally spoken: can you say something about the sensitivity of the single components in Figure 1?

The authors consider that climatic and non-climatic drivers can have an influence in land use and, at a certain point, in roughness changes. In turn, these factors can affect the evolution of inundation floods downstream the dam (Bornschein and Pohl, 2018; De Roo et al., 2001).

Although we cannot a priori state that the influence of a specific factor of the risk model will or will not be negligible for the final risk assessment, what we can say is that it is not always advisable to perform a risk analysis with a maximum level of detail. In some cases, it is convenient to include only those that will induce a significant effect on the risk, which would help avoiding unnecessary work or

Indeed, sensitivity studies must be carried out to understand the influence of the components in the risk model approach, as mentioned in your comment. Regarding this, some works have been published (Chauhan and Bowles, 2018; Escuder-Bueno et al., 2017) although in general further advances in this field can still be done, in particular when considering the additional uncertainties introduced by climate change.

**REFERENCES**

Bornschein, A. and Pohl, R.: Land use influence on flood routing and retention from the viewpoint of hydromechanics: Land use influence on flood routing and retention, Journal of Flood Risk Management, 11(1), 6–14, doi:10.1111/jfr3.12289, 2018.

Chauhan, S. and Bowles, D.: DAM SAFETY RISK ASSESSMENT WITH UNCERTAINTY ANALYSIS, 2018.

De Roo, A., Odijk, M., Schmuck, G., Koster, E. and Lucieer, A.: Assessing the effects of land use changes on floods in the meuse and oder catchment, Physics and Chemistry of the Earth, Part B: Hydrology, Oceans and Atmosphere, 26(7–8), 593–599, doi:10.1016/S1464-1909(01)00054-5, 2001.

Escuder-Bueno, I., Morales-Torres, A. and Castillo-Rodriguez, J. T.: Use of quantitative risk results to inform dam safety governance: Practical cases in Europe, in 36th Annual USSD Conference 2016, Denver, Colorado, USA., 2016.

Escuder-Bueno, I., Morales-Torres, A. and Castillo-Rodriguez, J.: EPISTEMIC VS NATURAL UNCERTAINTY IN DAM SAFETY DECISION MAKING: IS IT FAIR PLAY?, in 37th Annual USSD Conference, United States Society on Dams, Anaheim, California., 2017.

Lee, B.-S. and You, G. J.-Y.: An assessment of long-term overtopping risk and optimal termination time of dam under climate change, Journal of Environmental Management, 121, 57–71, doi:10.1016/j.jenvman.2013.02.025, 2013.

Morales-Torres, A., Serrano-Lombillo, A., Escuder-Bueno, I. and Altarejos-García, L.: The suitability of risk reduction indicators to inform dam safety management, Structure and Infrastructure Engineering, 1–12, doi:10.1080/15732479.2015.1136830, 2016.

Serrano-Lombillo, A., Morales-Torres, A., Escuder-Bueno, I. and Altarejos-García, L.: A new risk reduction indicator for dam safety management combining efficiency and equity principles, Structure and Infrastructure Engineering, 13(9), 1157–1166, doi:10.1080/15732479.2016.1245762, 2017.

Setrakian-Melgonian, M., Escuder-Bueno, I., Castillo-Rodriguez, J. T., Morales-Torres, A. and simarro-Rey, D.: Quantitative Risk Analysis to inform safety investments in Jaime Ozores Dam (Spain), International Commission of Large Dams (ICOLD), Prague, Czech Republic., 2017.

SPANCOLD: Risk Analysis as Applied to Dam Safety. Technical Guide on Operation of Dams and Reservoirs, Professional Association of Civil Engineers. Spanish National Committe on Large Dams, Madrid. [online] Available from: http://www.spancold.es/Archivos/Monograph_Risk_Analysis.pdf, 2012.

USBR: Chapter 3: General Spillway Design Considerations, in Appurtenant Structures for Dams (Spillways and Outlet Works), U.S. Bureau of Reclamation. [online] Available from: https://www.usbr.gov/tsc/techreferences/designstandards-datacollectionguides/designstandards.html, 2014.